# Mediterranean Diet and Genetic Determinants of Obesity and Metabolic Syndrome in European Children and Adolescents

**DOI:** 10.3390/genes13030420

**Published:** 2022-02-25

**Authors:** Miguel Seral-Cortes, Alicia Larruy-García, Pilar De Miguel-Etayo, Idoia Labayen, Luis A. Moreno

**Affiliations:** 1Growth, Exercise, NUtrition and Development (GENUD) Research Group, Faculty of Health Sciences, Instituto Agroalimentario de Aragón (IA2), Instituto de Investigación Sanitaria Aragón (IIS Aragón), Universidad de Zaragoza, 50009 Zaragoza, Spain; mseral@unizar.es (M.S.-C.); alarruy@unizar.es (A.L.-G.); lmoreno@unizar.es (L.A.M.); 2CIBER Fisiopatología de la Obesidad y Nutrición (CIBERobn), Instituto de Salud Carlos III, 28029 Madrid, Spain; 3Department of Health Sciences, Public University of Navarra, 31006 Pamplona, Spain; idoia.labayen@unavarra.es

**Keywords:** genetic risk score, single nucleotide polymorphism, interaction effect, Mediterranean Diet, obesity, metabolic syndrome, children and adolescents

## Abstract

Childhood obesity and metabolic syndrome (MetS) are multifactorial diseases influenced by genetic and environmental factors. The Mediterranean Diet (MD) seems to modulate the genetic predisposition to obesity or MetS in European adults. The FTO gene has also been shown to have an impact on the MD benefits to avoid obesity or MetS. Since these interaction effects have been scarcely analyzed in European youth, the aim was to describe the gene–MD interplay, analyzing the impact of the genetic factors to reduce the obesity and MetS risk through MD adherence, and the MD impact in the obesity and MetS genetic profile. From the limited evidence on gene–MD interaction studies in European youth, a study showed that the influence of high MD adherence on adiposity and MetS was only observed with a limited number of risk alleles; the gene–MD interplay showed sex-specific differences, being higher in females. Most results analyzed in European adults elucidate that, the relationship between MD adherence and both obesity and MetS risk, could be modulated by obesity genetic variants and vice versa. Further research is needed, to better understand the inter-individual differences in the association between MD and body composition, and the integration of omics and personalized nutrition considering MD.

## 1. Introduction

Obesity is defined as excess of body fat, and its prevalence has alarmingly increased over the last decades, with negative implications for the population’s health status [1]. In 2016, more than 6% of children and adolescents had obesity worldwide, with similar numbers in males and females [2]. Nowadays, the expected incidence of childhood obesity could be negatively escalating due to the COVID-19 pandemic [3,4]. Childhood overweight increases the risk of persistent obesity and related cardiometabolic events in adulthood [5]. Together with obesity, metabolic syndrome (MetS) is known to be a major health challenge in youth with increasing prevalence and a high risk of developing cardiovascular diseases in adulthood [6,7]. Both diseases could coexist within a given individual, acting in negative synergy to be fully established in a permanent status later in life [8]. The definition of MetS for children and adolescents comprises a number of cardiometabolic risk factors, such as total and/or central adiposity, dyslipidemia (high triglycerides and low high density lipoprotein (HDL)-cholesterol concentrations), hypertension, and insulin resistance [9]. Since obesity and MetS are complex and multifactorial diseases, they are influenced by genetic and environmental factors [10,11]. It seems that these factors do not act independently [12]. Instead, they either interact [13,14] or mediate [15,16] with each other to influence overweight and obesity risk.

In adults, the benefits of a high adherence to the Mediterranean Diet (MD) to prevent cardiovascular events are widely known [17]. It improves the lipid profile and adiposity levels [18], reduces the risk of overweight and obesity [19], MetS [18], type 2 diabetes mellitus (T2DM) [20] and hypertension [21], among others. Although less studied, the MD adherence is also relevant from the early stages in life. Breastfeeding could protect against childhood obesity [22], so maternal dietary habits could have an impact in the milk composition, thus, a balanced nutrition for the infant [23]. In fact, a study showed that breast milk mineral concentrations could be positively influenced by maternal MD adherence [24]. Moreover, in young age populations, MD has been observed to have inverse associations with obesity and MetS indicators such as high body mass index (BMI) [25], increased waist circumference (WC) [26], insulin resistance and high lipid profile [27]. As it is rather challenging to measure a whole diet [28], several diet scores assessing the degree of adherence to a certain dietary pattern have been developed to estimate their impact in a given population [29]. In this sense, the MD adherence has been extensively studied [30] through item-scales in different population groups [31,32].

Interestingly, the MD has previously shown to attenuate the genetic predisposition to obesity or MetS in European adults [33,34]. Individual single nucleotide polymorphisms (SNPs) or a combination of a number of SNPs have been considered as useful genetic tools to predict the predisposition to obesity and MetS in different age groups [35,36]. It is highly relevant studying gene-MD interaction effects from early stages in life, knowing that the development and implementation of chronic diseases could happen from preschool age due to low adherence to MD [37]. In European youth, the individual’s genetic profile has been previously observed to modulate the effect of MD in terms of obesity and MetS [38]. Although different approaches have previously considered MD as the dietary factor associated with reduction of obesity or MetS in children and adolescents [39,40], gene-MD interaction effects have scarcely been examined in youth. Therefore, the aims of the present study are: (i) to provide an overview of the MD benefits and its impact on body composition in European children and adolescents; (ii) to describe the potential gene-MD interaction effects that could modulate the risk of obesity and MetS development in European youth. Regarding the gene–MD interplay, we intend to analyze both interaction pathways, the impact of the genetic factors in the ability of MD to reduce the obesity and MetS risk, and to observe the MD impact on the genetic predisposition to obesity and MetS in European youth (Figure 1).

## 2. Obesity and Metabolic Syndrome: Definitions and Reference Values for Risk Estimation

To identify obesity in children and adolescents, the most commonly used methods are those based on anthropometric measurements. On one hand, single measurements such as triceps skinfold and waist circumference (WC) can predict total fat content in youth [41]. However, BMI is the most widely used anthropometric index. The World Obesity Federation (former International Obesity Task Force (IOTF)) BMI cut-off points are widely used to assess the prevalence of overweight and obesity in children [42]. Defining international cut-offs provides numerous advantages and it allows researchers to make comparisons with other BMI references [43]. However, the lack of uniformity in the methodological criteria to develop growth references for childhood BMI could sometimes lead to disparity in the results [44]. For instance, the World Health Organization’s (WHO) BMI references values might not me suitable for all European children [44]. Other BMI thresholds have been proposed by the U.S. Centers for Disease Control and Prevention (CDC), although the differences in the predictive capacity between the CDC reference and the World Obesity Federation’s reference are minimal [45]. A study of school aged children and adolescents showed a comparison of obesity estimations based on different reference values (CDC, IOTF and WHO criteria). The results showed the highest overweight and obesity underestimation in the WHO reference (8.97% and 5.67% respectively) [46]. More so, in a large cohort of Mediterranean children and adolescents, different BMI classification systems were assessed. The World Obesity Federation’s threshold showed higher specificity in assessing overweight and obesity whereas the WHO values had the highest sensitivity among all considered references [47]. As mentioned, obesity could also be assessed considering WC as a measurement of abdominal fatness estimation [48], since BMI could underestimate central obesity prevalence in young populations [49].

MetS comprises a cluster of anthropometric and biological markers which could include triglycerides, HDL cholesterol, systolic and diastolic blood pressure, waist circumference and homeostatic model assessment (HOMA) levels or fasting glucose, including obesity as a precondition of MetS development in children and adolescents [50]. The selection of the corresponding components depends on the elaboration of each MetS definition. A number of MetS definitions have been proposed throughout the years. The most widely used are the ones suggested by the WHO, National Cholesterol Education Program’s Adult Treatment Panel III (NCEP ATP III) and International Diabetes Federation (IDF), all to be thereafter adjusted versions to be applied in children and adolescents [9,51,52,53]. Different reference values have been displayed to develop MetS risk scores in children and adolescents [54,55]. This unified strategy increases the possibility to estimate and compare the prevalence and trends of cardiometabolic risk in youth when using continuous cardiometabolic risk scores.

## 3. Mediterranean Diet Assessment

Previous studies have shown a significant reduction of mortality associated with a greater adherence to the traditional MD, which considers both an adequate dietary pattern based on Mediterranean foods/products and a physically active lifestyle [56]. Trichopoulou, A. et al. observed inverse associations with greater MD adherence for death due to coronary heart disease and cancer in the adult population of Greece during a median of 44 month follow up [56]. The PREDIMED study in Spain, a multicenter randomized controlled trial of 4.8 years follow up, assessed the long term effects of the MD in terms of cardiovascular disease in adults, through different diet interventions supplemented with extra virgin olive oil or nuts. The study provided strong evidence that the MD could be an optimal dietary model for the management of cardiovascular events [57]. Nowadays, the traditional Mediterranean lifestyle is still fairly followed in the elderly, with some variability in this age group [58]. However, younger populations are moving away from the MD pattern/adherence [59]. Following a MD pattern involves the intake of extra virgin olive oil, high consumption of nuts and legumes, unrefined cereals, fruits and vegetables, moderate consumption of dairy products, fish and low consumption of meat and meat products [56]. The current understanding of MD is not only explained by the foods constituting the dietary pattern, but also on the whole process encompassing food production to food consumption (harvesting process), food consumption characteristics (seasonally and locally consumed), cooking techniques (extended use of olive oil with different condiments), eating behaviors (eating socially), daily physical activity and the impact that all these aspects might have on the overall health status [30].

In order to measure the degree of the adherence to the MD, different MD scores have been constructed to assess how food affects health [60]. One of the first and most extendedly used MD scores of all time was suggested by Trichopoulou, A. et al. [61]. Further adapted versions for children [62] and adolescents [63] were progressively emerging, resulting in useful tools to evaluate MD adherence in younger population. The scores were formed based on a number of typically consumed MD food groups, which varied from one another/from their recommendations. The information collected to build the mentioned indexes was performed through food frequency questionnaires (FFQ) [64] or 24 h recalls [65]. In order to calculate individual usual dietary intake, different correction methods could be used to adjust for between and within individuals’ variability [66,67]. These methods allow us to better estimate the individual’s adherence to MD based on the information provided.

## 4. Genomics of Obesity and Metabolic Syndrome

The development of overweight or obesity is the result of coexisting complex health determinants and their interactions at different levels. The genetic factors influencing the development of obesity have been consistently studied for the last decades [68]. The constant progress in genetic studies have extended the number of genetic variants, single nucleotide polymorphisms (SNPs), associated with BMI to 751 [69]. The *FTO* gene, known to be the first genome-wide association study (GWAS)-identified obesity gene [70], has previously been shown to have a relationship with obesity and metabolic pathways in European children [71] and adults [72]. Nevertheless, those SNPs only explain 6% of BMI variance. Recently, the use of a polygenic risk score (PRS) incorporating all information from 2.1 million SNPs regardless of their genome-wide significance, increased the variance explained by BMI to 23% [73]. Thus, the emphasis is now to understand the underlying mechanisms by which obesity related SNPs could influence body composition parameters. The mentioned PRS proposed by Khera et al. was based on adult BMI, but it also shows reliable associations in children [73]. Generally, the majority of GWAS loci for obesity related outcomes were identified in adult population. However, most of these mentioned loci are also associated with obesity in children and adolescents, suggesting that the genetic profile for obesity remains constant during the life course [74]. Interestingly, a study showed that adults BMI-related genetic variants were more significantly associated with child BMI during their adiposity rebound (around 5 years old), than in their adiposity peak (below 9 months old), with genetic variants slightly influencing BMI in the latter [75]. These findings strength the idea of the adult-based PRS use in children and adolescents. However, the use of a specific PRS based on children’s BMI could even increase the percentage of variance explained. Hence, the development of PRSs and genetic risk scores (GRS) (wide-genome significant) in early age populations would help to understand the expression mechanisms of obesity predisposition during childhood [76].

In the same way, MetS genetic susceptibility pathways have been evaluated. It was estimated that MetS heritability from European ancestry ranges between 10% and 30% [77,78]. In addition, numerous SNPs associated with individual MetS components have been reported in GWAS studies in different ethnic populations [79,80]. However, the effect of these SNPs on MetS with all included components remains understudied. Moreover, the majority of GWAS studies have been conducted in adult populations [81,82]. Nevertheless, understanding childhood obesity and MetS genetic pathways simultaneously could lead to observing partial overlapping, as sometimes genetic predisposition to MetS could be driven by obesity genes [83].

As mentioned before, GRSs might be fundamental for identifying individuals to be at high risk of obesity and/or MetS development [84]. In order to combine a number of SNPs to form a GRS, different methodologies can be followed: by summing the number of risk alleles we obtain an unweighted GRS (uGRS); by multiplying the number of risk alleles to each estimated coefficient, a weighted GRS (wGRS) is derived [36]. Although PRSs could contribute in a greater way to explain genetic variance to be at risk of a certain disease, a combination of genome-wide associated SNPs in a GRS form could contribute, a priori, to adding as much precision in the predictive ability to assess the risk of obesity and MetS as non-genome wide significant approaches (PRS), for clinical utility [85]. In any case, both methodologies have been shown to be strong and useful genetic tools to predict obesity and MetS genetic susceptibility in children and adolescents [73,76,83].

## 5. Association between Mediterranean Diet and Body Composition and Metabolic Syndrome in Youth

When evaluating MD dietary pattern/adherence in terms of obesity and MetS, it has been observed that MD adherence varies between Mediterranean countries within young population while little evidence is available in non-Mediterranean countries [32]. Thus, some non-interventional MD-obesity associations are likely to be conditioned by a generally poor MD adherence among youth [59]. Focusing our scope on European children and adolescents, diverse associations between MD and obesity related outcomes have been found in the literature. Higher adherence to the MD pattern was significantly associated with lower weight status [86], BMI [87], WC [88] and body fat percentage (%BF) [64].

Although less studied, the relationship between MD and MetS was also explored in youth. A high MD adherence resulted in glucose and lipid profile improvement after a lifestyle intervention [27] as well as a lower risk of overall MetS components [39]. In contrast, low MD adherence could contribute to higher central obesity, hypertriglyceridemia and insulin resistance [89]. However, not all studies considering MD as a healthy dietary pattern reported positive outcomes in terms of obesity and MetS in youth [32]. The inclusion of other Mediterranean lifestyle variables, such as physical activity, could complement the ineffective strategies following a dietary intervention alone, as it has been shown to play a key role on shaping the obesity [90] and cardiometabolic phenotype [91], more so when obesity is already established [92].

## 6. The Gene-Mediterranean Diet Interaction Effect in Obesity and Metabolic Syndrome

As it was the main focus of this narrative review, the present chapter is based on a systematic literature search considering gene–MD interaction effects and its relationship with body weight and composition in European children and adolescents. The focus was on assessing all studies considering the gene–MD interaction effects and its relationship with changes in body composition and biological parameters: BMI, WC or any of the MetS components (HOMA or glucose levels, HDL/Total cholesterol, systolic or diastolic BP) in children and adolescents. PUBMED was the electronic database searched. Mesh^®^ terms were used during the search strategy, based on medical subject headings and text words of peer papers identified. The search terms and words used were combined as follows: (“Obesity”[MeSH Terms] OR “Pediatric obesity”[MeSH Terms] OR “Metabolic syndrome”[MeSH Terms]) AND (diet, Mediterranean[MeSH Terms] OR “Diet score”[All Fields] OR “Diet indices”[All Fields]) AND (“Environmental exposure”[MeSH Terms] OR “Genetic predisposition to disease”[MeSH Terms]) AND (“child*”[Title/Abstract] OR “child”[MeSH Terms] OR “Preschool*”[Title/Abstract] OR “Child, Preschool*”[MeSH Terms] OR “adolescen*”[Title/Abstract] OR “adolescent”[MeSH Terms] OR “Youth”[Title/Abstract] OR “Teen*”[Title/Abstract] OR “Young people”[Title/Abstract]) AND (humans[Filter]). The following terms were also considered to screen genomic related studies: (“Genetic susceptibility”[MeSH Terms]) OR “Single nucleotide polymorphism”[MeSH Terms]) OR “genetic screening”[MeSH Terms]) OR “gene”[MeSH Terms]) OR “nutrigenomics”[MeSH Terms]) OR “nutrigenetics”[MeSH Terms]) OR “genetic interaction[MeSH Terms]”, although no results were found adding these terms to the intended search criteria. Additional filters were applied: written in the English language, population based-studies from Europe or individuals of European ancestry and age range from birth to 18 years old within targeted individuals. A total of three articles were obtained with the mentioned inclusion criteria, although after proofreading, only one article met the inclusion criteria proposed later in this this review [38]. The reference lists of all included manuscripts were double checked in order to identify potential missing articles that could have been ignored through the initial search.

This systematic search could set a milestone for a future in depth systematic review with the same established criteria when studies considering genomics and its impact on the benefits of MD among European youth will be more commonly found in the literature. The main findings of the identified study showed that the influence of high MD adherence on adiposity and MetS was only observed if a limited number of risk alleles were present. In addition, the gene–MD interaction effect showed sex-specific differences, being higher in females than in males (Figure 2). The analysis was carried out under the HELENA study, a cross-sectional multicentric study in European adolescents [93]. A cohort of 605 individuals aged 12.5–17.5 years old was used. To assess the MD adherence, a nine single component MD score was used [56,92], collected from 24 h recall questionnaires. Obesity-specific GRSs were developed in order to measure the genetic predisposition to adiposity and MetS [76]. The interaction effect resulted in being significantly associated to the main outcomes in both sex groups (*p* < 0.05).

No further studies assessing gene–MD interaction effects in terms of body composition or MetS in European children and adolescents were found. However, similar approaches of gene–MD interplay were observed in adult populations of European origin in relation to obesity and MetS. The studies shown in Table 1 and Table 2 were obtained as a result of removing the age filter and opening the search to all age categories. After selection of candidate articles, a total of 16 studies in European adult populations were considered; they were published from 2009 to 2021. Table 1 provides an overview of the impact of the genetic factors to influence the association between MD and obesity and MetS risk while Table 2 shows some examples observing the MD impact on the genetic predisposition to obesity and MetS. The assessed studies in European adults have considered different genetic approaches in order to evaluate the predisposition to obesity and MetS. Individual SNPs have predominantly been used as interacting genetic factor, to assess potential differences between individuals carrying different risk alleles. The *FTO*rs9939609 was the most common SNP considered in the obesity and MetS development [33,94,95,96,97]. Other previously obesity related SNPs were also included in other studies, such as *TCF7L2*rs7903146 [34,94] and *MC4R*rs17782313 [33,97]. In addition, the combined effect of a number of SNPs comprising a GRS was also considered in the gene–MD interaction analyses. The GRSs observed in the studies analyzed were formed from 2 [11] to 77 obesity related SNPs [98]. The mentioned GRS formed by 77 SNPs, was built from loci associated to BMI in adults of European descent. A weighted method was applied to calculate the GRS on the basis of the selected SNP’s relative effect size. The GRS ranged from 0–154, with each unit corresponding to one risk allele and higher GRS indicating higher predisposition to obesity [98]. This GRS was applied in a large combined prospective cohort (*N* = 14,046) of individuals of European ancestry during a 20 year follow up, where it was observed that the benefits of an improved diet quality were more pronounced in those individuals at higher genetic risk of obesity [98]. As this manuscript considered both genes and MD as modulating factors, it was suitable to be included in duplicate, so further description of this study’s characteristics can be found in Table 1 and Table 2. In this sense, the description of the main outcomes and the statistical models analyzing the gene–diet interplay were interpreted, in order to better understand the strategy used to assess the interaction effect in each studied article.

**Table 1 genes-13-00420-t001:** Studies considering the individuals genetic profile to modulate the benefits of MD in relation to obesity and MetS in European adults. Articles ordered by outcome (starting with obesity, then MetS) and publication year (starting by most recent).

Author	Outcome	Year	Country	Age Group	Study Design	Sample Size	Diet Assessment ^†^	Genetic Input	Results
Wang, T. et al. [98]	Obesity	2018	US (European ancestry)	Adults. 30–55 yo	Prospective cohort study. 20 year follow-up	14,046 (62.8% females)	FFQ. Traditional MDS (9-point score)	77 obesity related SNP GRS	The beneficial effect of improved diet quality on weight management was particularly pronounced in people at high genetic risk for obesity.
Roswall, N. et al. [94]	Obesity	2014	Multicentric: 5 European countries.	Adults. 47.6 ± 7.5 yo	Longitudinal. Median follow-up 6.8 years	11,048 (55.1% females)	FFQ. Traditional MDS (18-point score)	2 SNPs: *FTO*rs9939609 and *TCF7L2*rs7903146	High MDS was associated with lower changes in WC and BMI, regardless of *FTO* and *TCF7L2* risk alleles. In weight, the effect may depend on the T*CF7L2*rs7903146 variant (beneficial effect only in subjects with 1 or 2 risk alleles).
Lowry, E. et al. [11]	MetS	2018	Canada (European ancestry)	Adults 60.7 ± 0.73 yo	Longitudinal. 1 year intervention	159 (51.6% females)	24 h recall. Mediterranean based Canadian Healthy eating Index. (Range 0–100) [99]	2 SNPs: *APOA5*rs662799 and *ADIPOQ*rs1501299 GRS	Participants carrying none of the risk alleles in the 2 SNPs (GRS = 0) showed the greatest reduction in MetS score during the intervention.

Legend. List of abbreviations: Apolipoprotein A5 (APOA5); Adiponectin (ADIPOQ); Body Mass Index (BMI); Food Frequency Questionnaire (FFQ); Fat Mass and Obesity-Associated (FTO); Genetic Risk Score (GRS); Mediterranean Diet Score (MDS); Metabolic Syndrome (MetS); Randomized clinical trial (RCT); Single Nucleotide Polymorphism (SNP); Transcription Factor 7 Like 2 (TCF7L2); United States (US); Waist Circumference (WC) and years old (yo). ^†^: Type of dietary questionnaire and type of MD score used were displayed. FFQ questionnaires were used in Wang, T. et al. and Roswall, N. et al. to measure the adherence to typically consumed items in the MD based on the traditional MDS recommendations [56]. Lowry, E. et al. used a 24 h recall to then consider the Canadian eating index, adapted to the Mediterranean lifestyle [100]. The higher the score obtained from the information provided in the questionnaire, the greater the adherence to the Mediterranean Diet. Applicable to all MD scores evaluated in the present table.

**Table 2 genes-13-00420-t002:** Studies considering the MD as interaction factor to modulate the association between the genetic risk and obesity or MetS related outcomes in European adults. Articles ordered by outcome (starting by obesity, then MetS) and publication year (starting by the most recent).

Author	Outcome	Year	Country	Age Group	Study Design	Sample Size	Diet Assessment ^†^	Genetic Input	Results
Baratali, L. et al. [101]	Obesity	2021	Switzerland	Adults. CS: 58.4 ± 10.6 yo PS: 58.0 ± 10.4 yo	Cross-sectional and prospective. 5.3 year follow-up	CS: 3033 PS: 2542 (CS: 53.2% females; PS: 54.7% females)	FFQ. 2 MDS: Traditional MDS I (8-point score) Swiss MDS II (9-point score)	2 obesity GRSs based on 31 and 68 SNPs.	No gene-diet interaction was found for changes in obesity markers, suggesting that diet exerts the same effect irrespective of the genetic background of the participants.
Wang, T. et al. [98]	Obesity	2018	US (European ancestry)	Adults. 30–55 yo	Prospective cohort study. 20 year follow-up	14,046 (62.8% females)	FFQ. Traditional MDS (9-point score)	77 obesity related SNP GRS	MD could not significantly attenuate the genetic association with increases in BMI and body weight.
Livingstone, K.M. et al. [95]	Obesity	2016	Multicentric. 7 European countries	Adults. 40.4 ± 13.0 yo	RCT. 6 month follow-up	1607 (58.0% females)	FFQ. PREDIMED MDS (14-point score)	1 SNP: *FTO*rs9939609	No evidence of interactions between *FTO* genotype and dietary intakes on BMI and WC were found.
Corella, D. et al. [102]	Obesity	2014	Spain	Adults. 67.0 ± 6.2 yo	RCT. Median follow up 4.8 years	7161 (57.4% females)	FFQ. PREDIMED MDS (14-point score)	1 SNP: *FAIM2*rs7138803	No statistically significant gene-diet interactions between MD and *FAIM2*rs7138803 were found in determining BMI.
Corella, D. et al. [33]	Obesity	2012	Spain	Adults. 67.0 ± 6.2 yo	RCT. Median follow up 4.8 years	7052 (57.3% females)	FFQ. PREDIMED MDS (14-point score)	2 SNPs: *MC4R*rs17782313 and *FTO*rs9939609	Statistical and biological interactions with MD modulate the effects of *FTO* and *MC4R* polymorphisms on obesity.
Razquin, C. et al. [96]	Obesity	2010	Spain	Adults. 55–80 yo	RCT. 3 year follow-up	776 (54.9% females)	FFQ. Not specified	1 SNP: *FTO*rs9939609	After a nutritional intervention with MD, A-allele carriers had lower body weight gain than wild type subjects.
Razquin, C. et al. [103]	Obesity	2010	Spain	Adults. 55–80 yo	RCT 3 year follow-up. MD + virgin olive oil	737 (55% females)	FFQ. Not specified	1 SNP: −174G/C on the *IL*6 gene	After a nutritional intervention with MD + olive oil, CC subjects for the −174G/C had the greatest reduction in body weight.
Razquin, C. et al. [104]	Obesity	2009	Spain	Adults. 55–80 yo	RCT 3 year follow-up MD + virgin olive oil	774 (59.3% females)	FFQ. Not specified	1 SNP: Pro12Ala of the *PPARγ*gene	After a nutritional intervention with MD, reduced WC was observed among the population studied, reversing the negative effect of the 12Ala allele carriers
Coltell, O. et al. [105]	MetS	2021	Spain	Adults 67.0 ± 0.2 yo	Cross-sectional	954 (63.5% females)	FFQ. Spanish short screener (14-point score)	GWAS identified *OPCML*rs2917570	If MD is low, the minor allele of the rs2917570 is associated with higher adiponectin concentration. However, when adherence to MD is high, the minor allele is associated with lower adiponectin concentration. *
Coltell, O. et al. [106]	MetS	2020	Spain	Adults 65.1 ± 0.2 yo	Cross-sectional	426 (56.5% females)	FFQ. PREDIMED PLUS MDS (17-point score)	13 SNPs from GWAS in the *ME*1 gene in relation to serum omega- 3 PUFA	When MD adherence to is low, the minor allele is associated with an increase in serum omega-3 PUFA concentrations. If MD adherence is high, the minor allele is associated with a decrease in serum omega-3 PUFA concentrations. **
San Cristobal, R. et al. [107]	MetS	2017	Multicentric. 7 European countries	Adults. 40.8 ± 13.0 yo	RCT. 6 month follow-up. On-line	1263 (57.1% females)	FFQ. PREDIMED MDS (14-point score)	14 SNPs GRS of MetS related traits	Higher GRS may reduce MD adherence benefits on total cholesterol concentration.
Corella, D. et al. [108]	MetS	2016	Spain	Adults. 66.9 ± 6.2 yo	RCT. Median follow up 4.8 years	7098 (58.2% females)	FFQ. PREDIMED MDS (14-point score)	1 SNP: *CLOCK*rs4580704	The interaction between the SNP and MD did not reach the statistical significance and the heterogeneity by MD is not confirmed.
Ortega-Azorín, C.S. et al. [109]	MetS	2014	Spain	Adults. 66.9 ± 6.2 yo	RCT. Median follow up 4.8 years	7166 (58.1% females)	FFQ. PREDIMED MDS (14-point score)	1 SNP: *MLXIPL*rs3812316	MD enhances the triglyceride lowering effect of the *MLXIPL*rs3812316 variant.
Corella, D. et al. [34]	MetS	2013	Spain	Adults. 67.0 ± 6.2 yo	RCT. Median follow up 4.8 years	7018 (57.4% females)	FFQ. PREDIMED MDS (14-point score)	1 SNP: TCF7L2rs7903146	MD may reduce increased fasting glucose and lipids in TT individuals.
Ortega-Azorín, C.S. et al. [97]	MetS	2012	Spain	Adults 66.9 ± 6.2 yo	RCT. Median follow up 4.8 years	7052 (57.3% females)	FFQ. PREDIMED MDS (14-point score)	2 SNPs: *FTO*rs9939609 and *MC4R*rs17782313	The *FTO*rs9939609 and the *MC4*R-rs17782313 association with T2DM depends on diet. High MD adherence counteracts the genetic predisposition.

Legend. List of abbreviations. Body Mass Index (BMI); Clock circadian regulator (CLOCK); Cross-sectional (CS); Fas Apoptotic Inhibitory Molecule 2 (FAIM2); Fat Mass and Obesity-Associated (FTO); Food Frequency Questionnaire (FFQ); Genetic Risk Score (GRS); Genome Wide Association Study (GWAS); Interleukin 6 (IL6); Malic Enzyme 1 (ME1); Mediterranean Diet (MD); Mediterranean Diet Score (MDS); Max-like protein X interacting protein-like (MLXIPL); Melanocortin-4 Receptor (MC4R); Opioid Binding Protein/Cell Adhesion Molecule Like (OPCML); Polyunsaturated fatty acid (PUFA); Prospective (PS); Randomized clinical trial (RCT); United States (US); Transcription factor 7-like 2 (TCF7L2); Type 2 Diabetes Mellitus (T2DM); Waist circumference (WC) and years old (yo). ^†^: Type of dietary questionnaire and type of MD score used were displayed. FFQ questionnaires were used in all cases to measure the adherence to typically consumed items in the MD. In terms of MDS, Baratalli et al. considered 2 MDS: one following the traditional recommendations (MDS I) [56] and another one considering dairy products as beneficial items (MDS II) [110]. Wang, T. et al. used traditional MDS recommendations [56]. Livingstone, K.M. et al., Corella, D. et al. (2012, 2013, 2014 and 2016), San Cristobal, R. et al. and Ortega-Azorín, C.S. et al. (2012 and 2014) manuscripts used the MDS estimation based on the adapted version of the PREDIMED study [17,57]. Dietary assessment in Razquin, C. et al. (2010) considered PREDIMED recommendations, although no measuring MDS was reported. Coltell, O. et al. (2020) considered the MDS developed in the PREDIMED-PLUS trial [111]. Coltell, O. et al. (2021) considered a validated short screener in Spanish population [112]. The higher the score obtained from the information provided in the questionnaire, the greater the adherence to the Mediterranean Diet. Applicable to all MD scores evaluated in the present table. * Adiponectin: possible protective role against insulin resistance and arteriosclerosis. ** Omega-3 PUFA: potential beneficial effects in cardiovascular and metabolic risk factors.

## 7. Evaluation of the Genomic Role in the Mediterranean Diet Impact: Children, Understudied Population

Although some authors have shown the ability of the MD and the genetic profile to interact, modulating the risk associated with obesity and MetS parameters in adults of European origin, almost no studies assessing gene–diet interaction effects have been identified in European youth. Therefore, the present review has shown that gene–diet interaction effects in early life remain deeply understudied in young individuals of European origin.

From the only study extracted from the targeted inclusion criteria of this review’s search, promising results were shown in terms of the influence of the individual’s genetic risk to obesity, acting as a modulating factor in the benefits that a high MD adherence exerts on European adolescents of the HELENA study. The obesity-GRS developed to predict the risk of adiposity and MetS in this cohort of European adolescents showed that when adherence of MD was high, certain individuals, based on their genotype, had lower BMI, WC or MetS score. On the other hand, a small fraction of the selected sample was not genetically predisposed to have better body composition or metabolic profile despite a high MD adherence, or they could even see their health status worsen at that point [38]. Another important finding in the mentioned study was the sex-specific differences observed when the genotype was influencing the relation to MD and adiposity and MetS, as the interaction effect was higher in females than in males [38]. Similarly, a study performed in European adults explored gene–MD and gene–sex interactions in terms of polyunsaturated fatty acids (PUFAs) levels, showing that sex could be another relevant factor explaining differences in the effect of genetics on PUFAs [106].

The importance of assessing gene–diet interactions in adults relies on the fact that the associations of childhood BMI with adult diseases are explained by shared genetic factors [113], meaning that a partial genetic overlap exists in the biological processes underlying children’s BMI with those underlying adults’ BMI and cardiometabolic traits [113]. However, it is also possible that this transition might be explained through phenotypic continuity of BMI from childhood into adulthood [113]. Since there is evidence of genetic correlations between child and adult BMI, some authors carried out a pleiotropy test (shared genetics) and functional enrichment of SNPs associated with childhood BMI and 15 adult cardiometabolic traits [114]. This study showed that pleiotropic genetic effects and enrichment of functional annotations in genetic variants were significantly associated with both childhood obesity and cardiometabolic diseases in adulthood.

In light of the above, a number of studies have been identified, observing how the genes and MD interplay act to modify body composition in relation to obesity and MetS in European adults. For instance, the previously mentioned PREDIMED study has contributed to disseminate a series of gene–MD interaction results in European adults to prevent cardiovascular events through an MD-based intervention [57]. The consumption of MD products was assessed through a 14-point scale extracted from FFQs when the greater the score, the greater the MD adherence estimation [112]. Additionally, different individual SNPs have been explored to assess whether the MD could modulate the genetic predisposition to certain components related to obesity or MetS. In this sense, not all the gene-MD interactions studied resulted in a significant effect where the primary outcome was improved by MD adherence. Instead, although some studies have shown positive interaction effects in terms of obesity [33] and MetS [34,98,110], no statistical significance was reached in other studies. Therefore, the MD adherence could not always be confirmed to be beneficial for obesity [102] and MetS [108] respectively. More so, GWAS studies in European adults have established novel associations between SNPs and metabolic variables where MD was acting as an interaction factor [106] whereas other known obesity related SNPs, such as *FTO*rs9939609, have also proved to reduce the weight gain of certain allele carriers of different cohorts after nutritional intervention [33,96]. However, no consistency was found in other multicentric studies in this regard [94,95].

A combination of SNPs in a GRS format was also considered in other analysis to predict the genetic risk of obesity and MetS in European adults. The studies evaluated showed diversity in their results. On one hand, significant interactions, suggesting that a certain genetic profile might predict an adverse response to what one would normally expect from a healthy eating pattern. On the other hand, non-significant ones [11,107] suggest otherwise [101]. One of the studies analyzed in the present review (Wang, T. et al.), used an obesity-GRS to detect individuals with a genetic predisposition to obesity in two prospective cohorts [98]. MD could not significantly attenuate the association between genetic susceptibility and increases in BMI and body weight, although improving adherence to the Alternate Healthy Eating Index 2010 (AHEI-2010) [115] or Dietary Approach to Stop Hypertension (DASH) [116] patterns could attenuate the genetic association with weight gain. At the same time, they assessed whether the beneficial effect of improved diet quality on weight management was particularly pronounced in individuals at high genetic risk of obesity. In this case, the beneficial effect of improved diet quality on weight management was particularly pronounced in people at high genetic risk for obesity. The role of the interaction factor was doubly considered, depending on whether the main goal was to observe the influence of MD on the genetic predisposition to obesity, or whether the individuals’ genotype could influence the use of health benefits from the MD to reduce the risk of obesity [117]. The role of MD attenuating the genetic susceptibility to obesity and MetS of certain individuals is the commonly found approach in gene–diet interaction studies in the literature [57]. However, the benefits of the adherence to a healthy diet, such as the MD, are not universally influenced in all possible genotypic profiles. Therefore, not all possible allele combinations would benefit from the same pattern. The mentioned study strengthens the idea of evaluating all possible interaction pathways at the same time, so the role of both, MD and genetic risk to a certain disease, acting as modulative factors, is fully understood.

Further studies considering the MD and genetic susceptibility as attenuating factor were also explored in studies which main outcome was partially related to our exposure. For instance, women carrying the T allele of the *TCF7L2*rs7903146 who also had high MD adherence early in pregnancy, had lower risk of developing gestational diabetes mellitus than CC carriers [118]. The same SNP-diet interaction was analyzed in terms of weight status [94] and fasting glucose and lipids [34] in European adults, where the effect of MD could depend on the *TCF7L2*rs7903146. Selected SNPs from GWAS were observed to interact with MD in relation to serum bilirubin concentrations (cardiovascular risk biomarker) [119]. At candidate gene level, *PPARγ2*rs1801282-MD interactions were observed to favor telomeres length (obesity and cardiovascular disease are linked to shortened telomeres) [120] and no MD effect was observed in terms of inflammation markers regardless of the inflammation related *COX-2* -765G>C and *IL-6* -174G>C polymorphisms [121].

In order to find relevant gene–diet interaction analysis in relation to obesity and MetS in European children and adolescents, we need to consider different dietary patterns or specific nutrient intakes other than MD. Such is the case of the α-linolenic acid dietary intake, which interacted with FADS1 genetic variability to affect serum non-HDL-cholesterol concentrations in European adolescents [122], or another study where the genetic liability to obesity, assessed by a obesity polygenic risk score (PRS), was attenuated by a higher fiber intake [123].

The present review showed some limitations. First, the targeted topic of the present review represented a small fraction in the literature, as the gene-MD interaction effect analysis in terms of body composition in European children and adolescents are currently under development. Second, due to the narrow scope of the systematic search, little evidence was found with the intended search criteria in young population. Therefore, the majority of the studies evaluated in the present review were conducted in European adults. Lastly, due to the consistently low adherence of the MD among different age populations, the results showed in the mentioned studies of this review should be interpreted carefully.

The optimal use of the benefits of MD for a more effective prevention and treatment programs of obesity and MetS in youth is through a personalized diet. In this sense, some intervention studies considering a personalized approach with MD have been put into practice in recent years. For example, the MED4youth study, an ongoing multicenter RCT aimed to reduce obesity levels through MD vs. a traditional low-fat diet in European adolescents [124]. One of the main goals of the study is to apply an omics approach to observe whether the proposed interventions could modulate gut microbiota and derived metabolites in order to investigate their mechanisms and maximize the beneficial effect of a high MD adherence. In addition, the Food4me study recruited European adults to perform an internet-based nutritional intervention to evaluate the effect of personalized interventions on dietary changes associated with MD [125]. When participants were randomly assigned to receive different kinds of intervention advice, the personalized nutrition advice at diet, phenotype and genotype level showed the largest differences in terms of MD score [125]. These types of tailored interventions highlight the relevance of understanding the molecular basis of the MD effect. The integration of post-GWAS, cross-disciplinary collaborations combining omics technologies and computational techniques enable us to incorporate a significant number of biological markers using genomics, epigenomics, metagenomics, metabolomics and so forth, which better reflect the interaction processes taking place at molecular and cellular levels [126,127,128,129]. These contributions so far have helped researchers to understand the MD effects on the intermediate or final phenotypes of cardiovascular health [130]. However, dealing not only with a single MD component, but with a combination of foods, nutrients and phytochemicals, can provide separate or combined effects at different levels. This rather complex and multidimensional task becomes the next challenge for the forthcoming years of research [131].

## 8. Conclusions

The present review confirms that the numerous approaches carried out considering MD as healthy dietary pattern had, in general, an association with low obesity and MetS frequencies in children and adolescents. The number of gene–MD interaction analyses in relation to the risk of obesity or MetS performed in young populations are considerably limited. However, the few results analyzed in European youth suggest the possibility that obesity related genotypes modulate the relationship between MD adherence and obesity and MetS risk. Further research is needed to better understand the inter-individual differences in the association between MD and obesity and MetS, as well as the mechanisms behind the protective effects of MD in the overall cardiovascular health through the integration of omics and a personalized nutrition approach considering MD. 

## Figures and Tables

**Figure 1 genes-13-00420-f001:**
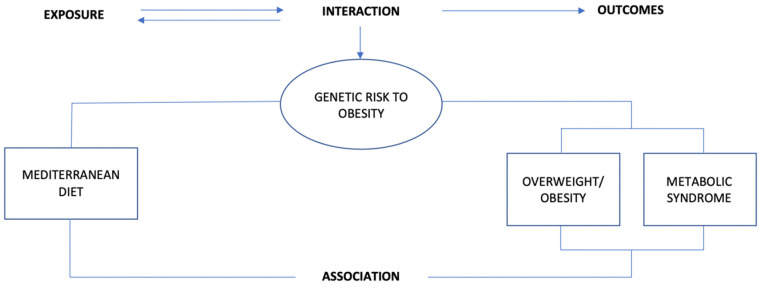
Conceptual framework of the present narrative review.

**Figure 2 genes-13-00420-f002:**
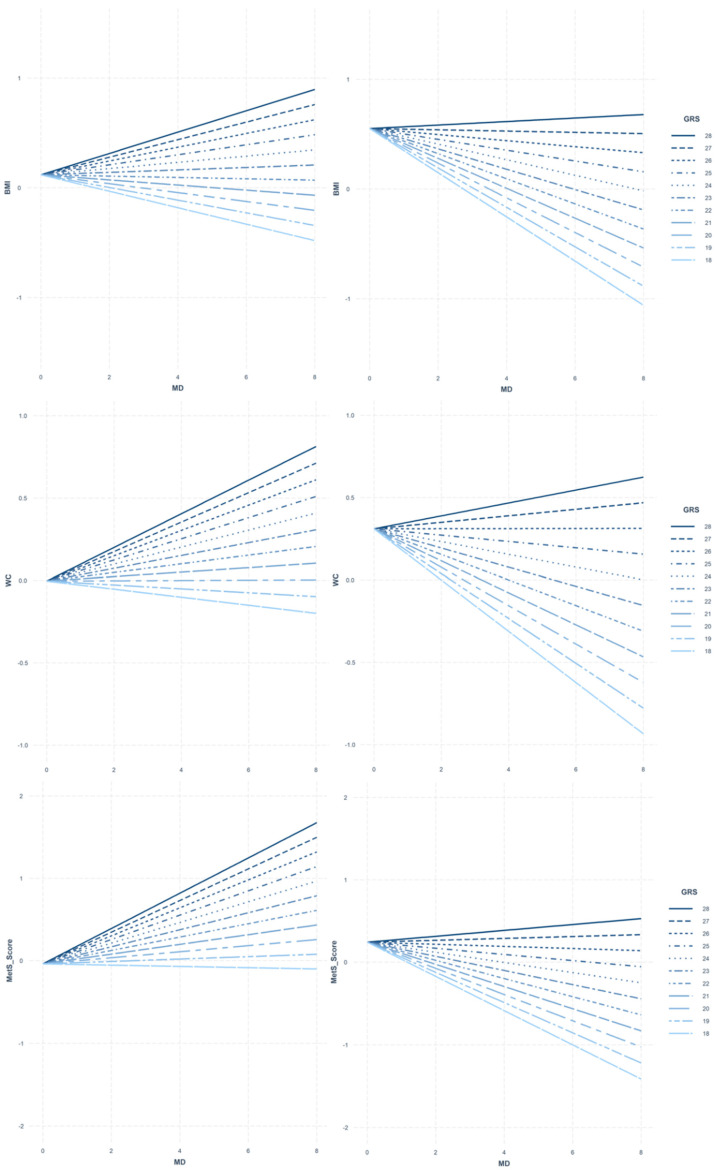
Interaction models between body mass index (BMI), waist circumference (WC), and metabolic syndrome score (MetS Score) and the Mediterranean diet (MD) according to the obesity genetic risk score (Obesity-GRS) modulation compared by gender (males left panel, females right panel). The Obesity-GRS values (18–28) are displayed according to our population distribution. Legend: The population distribution design follows different line patterns with reference points to observe the trend of the adolescent cohort according to the genetic risk for obesity. To analyze these results represented in the figure, a positive gradient shows the MD acting as arisk factor, while a negative gradient shows the protective role of the MD.

## Data Availability

Not applicable.

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
