# Peer review of "Mediterranean Diet and Genetic Determinants of Obesity and Metabolic Syndrome in European Children and Adolescents"

_genes, 2022, doi:10.3390/genes13030420_

Round 1

Reviewer 1 Report

Thank you for allowing me to review this manuscript entitled "Mediterranean diet and genetic determinants of obesity and metabolic syndrome in European children and adolescents."
The featured article provides concise and appropriate updates on the latest advances in childhood obesity and metabolic syndrome (MetS), as multifactorial diseases influenced by genetic and environmental factors.
It is an interesting project, with a very current theme. However, it has several limitations that make it suitable for publication in this journal. These limitations are detailed below:

In the introduction section:
I would recommend that the authors justify the novelty and relevance of the study being conducted in more detail.
Lines 127-130: It is specified that previous studies have shown a significant reduction in mortality associated with greater adherence to traditional DM, which considers both an adequate dietary pattern based on Mediterranean foods/products and a physically active lifestyle. I would recommend justifying and expanding this information: Since when is this reduction evident? How does it affect health? Are there differences between European countries?
On the other hand, I would recommend that the authors make a paragraph of limitations, specifying the weak points of this manuscript.
In relation to the conclusions, they are adequate, clear and precise, but it would be important to add the lines of the future that the authors consider.
good job

Reviewer 2 Report

Reviewer’s comment

In the review article titled ‘Mediterranean Diet and Genetic determinants of obesity and metabolic syndrome in European children and adolescents', the authors Miguel et al. have reviewed the published information on the influence of Mediterranean diet (MD) on genetic determinants of obesity and metabolic syndrome. The authors want to convey that the impact of MD on genetic factors of obesity children/youth to be studied further, especially in Europe as most of the studies conducted were on adults. The aim of the article is important, especially when there are increasing cases of childhood obesity in many parts of the world, including Europe. The article is well written but can be more reader-friendly. Following are a few suggestions that might increase the value of the manuscript.

  1. Abstract: Looks too general. As the article title has ‘genetic determinants,’ the authors may consider including the importance of genes like FTO related to metabolic syndrome and obesity in the abstract. A modified version of lines 243-246 (The main finding….than in males) can be included in the abstract and conclusion.
  2. Abstract: Line 27-28. ‘One study showed that the genetic profile was influencing the benefits of a high MD adherence in terms of body composition. Precisely, what is the influence? 
  3. Figure1: The authors may consider modifying Fig 1 such that it can be grasped quickly for a wide range of readers. The arrow mark usually represents an increase. An arrow mark from ‘Mediterranean diet’ to ‘Genetic risk to the obesity’ indicates the MD increases the Genetic risk. 
  4. Line153-193. Most of the genetic factors of adult obesity are also common to children (Loos and Yeo 2022, Nature reviews genetics). As there are a smaller number of studies on childhood obesity, the authors may consider including a paragraph on adult genetic factors  
  5. The tables are a good source of information. However, although the authors have cited references, it is good to include the contents and describe them in detail. Going back to all the back-references may not be reader-friendly. 
  6. Line 41: Recent data can be included (Loos and Yeo 2022, Nature reviews genetics) if not authors want to restrict it to 2009 to 2021.
  7. From the definition, the MD components look like it is for adult or grown up children. Milk or milk products are not preferred in MD, but breast milk is generally highly encouraged for children. The WHO report indicates breastmilk can reduce childhood obesity (https://www.euro.who.int/en/health-topics/noncommunicable-diseases/obesity/news/news/2019/4/new-who-studies-europe-battles-childhood-obesity-and-experts-confirm-breastfeeding-protects-against-child-obesity). Moro et al. conducted studies in Italy on the composition of breast milk of pregnant women who are on MD (https://www.frontiersin.org/articles/10.3389/fped.2019.00066/full). The authors may consider including some of these related studies in the introduction, although not related to genetic factors. 

8. Line 100-101 (WHO) ‘BMI references values are lower in early life and higher in later life compared to other references, as the methods applied to follow different growth standards [38]’ - The authors may consider including some reference values
